# META-LEARNING NEURAL BLOOM FILTERS

## ABSTRACT

There has been a recent trend in training neural networks to replace data structures that have been crafted by hand, with an aim for faster execution, better accuracy, or greater compression. In this setting, a neural data structure is instantiated by training a network over many epochs of its inputs until convergence. In many applications this expensive initialization is not practical, for example streaming algorithms — where inputs are ephemeral and can only be inspected a small number of times. In this paper we explore the learning of approximate set membership over a stream of data in one-shot via meta-learning. We propose a novel memory architecture, the Neural Bloom Filter, which we show can be more compressive than Bloom Filters and several existing memory-augmented neural networks.

## 1 INTRODUCTION

One of the simplest questions one can ask of a set of data is whether or not a given query is contained within it. Is $q$, our query, a member of $S$, our chosen set of observations? This *set membership* query arises across many computing domains; from databases, network routing, and firewalls. One could query set membership by storing $S$ in its entirety and comparing $q$ against each element. However, more space-efficient solutions exist.

The original and most widely implemented *approximate set membership* data-structure is the Bloom Filter (Bloom, 1970). It works by storing sparse distributed codes, produced from randomized hash functions, within a binary vector. The Bloom-filter trades off space for an allowed false positive rate, which arises due to hash collisions. However its error is one-sided; if an element $q$ is contained in $S$ then it will always be recognized. It never emits false negatives.

One can find Bloom Filters embedded within a wide range of production systems; from *network security* (Geravand & Ahmadi, 2013), to block malicious IP addresses; *databases*, such as Google's Bigtable (Chang et al., 2008), to avoid unnecessary disk lookups; *cryptocurrency* (Hearn & Corallo, 2012), to allow clients to filter irrelevant transactions; *search*, such as Facebook's typeahead search (Adams, 2010), to filter pages which do not contain query prefixes; and *program verification* (Dillinger & Manolios, 2004), to avoid recomputation over previously observed states.

While the main appeal of Bloom Filters is favourable compression, another important quality is the support for dynamic updates. New elements can be inserted in $\mathcal{O}(1)$ time. This is not the case for all approximate set membership data structures. For example, perfect hashing saves $\approx 40\%$ space over Bloom Filters but requires a pre-processing stage that is polynomial-time in the number of elements to store (Dietzfelbinger & Pagh, 2008). Whilst the static set membership problem is interesting, it limits the applicability of the algorithm. For example, in a database application that is serving a high throughput of write operations, it may be intractable to regenerate the full data-structure upon each batch of writes.

We thus focus on the data stream computation model (Muthukrishnan et al., 2005), where input observations are assumed to be ephemeral and can only be inspected a constant number of times — usually once. This captures many real-world applications: network traffic analysis, database query serving, and reinforcement learning in complex domains. Devising an approximate set membership data structure that is not only more compressive than Bloom Filters, but can be applied to either dynamic or static sets, could have a significant performance impact on modern computing applications. In this paper we investigate this problem using memory-augmented neural networks and meta-learning.

We build upon the recently growing literature on using neural networks to replace algorithms that are configured by heuristics, or do not take advantage of the data distribution. For example, Bloom Filters are indifferent to the data distribution. They have near-optimal space efficiency when data is drawn uniformly from a universe set (Carter et al., 1978) (maximal-entropy case) but (as we shall show) are sub-optimal when there is more structure. Prior studies on this theme have investigated compiler optimization (Cummins et al., 2017), computation graph placement (Mirhoseini et al., 2017), and data index structures such as b-trees (Kraska et al., 2018).

In the latter work, Kraska et al. (2018) explicitly consider the problem of static set membership. By training a neural network over a fixed $S$ (URLs from Google's Transparency Report) with negative examples in the form of held-out URLs, they observe $36\%$ space reduction over a conventional Bloom Filter[1]. Crucially this requires iterating over the storage set $S$ a large number of times to embed its salient information into the weights of a neural network classifier. For a new $S$ this process would have to be repeated from scratch.

Instead of learning from scratch, we draw inspiration from the few-shot learning advances obtained by meta-learning memory-augmented neural networks (Santoro et al., 2016; Vinyals et al., 2016). In this setup, tasks are sampled from a common distribution and a network learns to specialize to (learn) a given task with few examples. This matches very well to applications where many Bloom Filters are instantiated over different subsets of a common data distribution. For example, a Bigtable database usually contains one Bloom Filter per SSTable file. For a large table that contains Petabytes of data, say, there can be over $100,000$ separate instantiated data-structures which share a common row key format and query distribution. Meta-learning allows us to exploit this common redundancy.

The main contributions of this paper are (1) A new sparse memory-augmented neural network architecture, the *Neural Bloom Filter*, which learns to write to memory using a distributed write scheme, and (2) An empirical evaluation of the Neural Bloom Filter meta-learned on one-shot approximate set membership problems of varying structure.

We compare with the classical Bloom Filter alongside other memory-augmented neural networks such as the Differentiable Neural Computer (Graves et al., 2016) and Memory Networks (Sukhbaatar et al., 2015). We find when there is no structure, that differentiates the query set elements and queries, the Neural Bloom Filter learns a solution similar to a Bloom Filter derivative (a Bloom-g filter (Qiao et al., 2011)), but when there is a lot of structure the solution can be considerably more space-efficient.

## 2 BACKGROUND

### 2.1 APPROXIMATE SET MEMBERSHIP

The problem of *exact set membership* is to state whether or not a given query $q$ belongs to a set of $n$ distinct observations $S = \{x_1, \ldots, x_n\}$ where $x_i$ are drawn from a universe set $U$. By counting the number of distinct subsets of size $n$ it can be shown that any such exact set membership tester requires at least $\log_2 \binom{|U|}{n}$ bits of space. To mitigate the space dependency on $|U|$, which can be prohibitively large, one can relax the constraint on perfect correctness. *Approximate set membership* allows for a false positive rate of at most $\epsilon$. Specifically we answer $q \in A(S)$ where $A(S) \supseteq S$ and $p(q \in A(S) - S) \leq \epsilon$. It can be shown[2] the space requirement for approximate set membership of uniformly sampled observations is at least $n \log_2(\frac{1}{\epsilon})$ bits (Carter et al., 1978) which can be achieved with perfect hashing. So for a false positive rate of $1\%$, say, this amounts to $6.6$ bits per element. In contrast to storing raw or compressed elements this can be a huge space saving, for example ImageNet images require $108$ KB per image on average when compressed with JPEG, an increase of over four orders of magnitude.

### 2.2 BLOOM FILTER

The Bloom Filter (Bloom, 1970) is a data structure which solves the dynamic approximate set membership problem with near-optimal space complexity. It assumes access to k uniform hash functions

---

[1]The space saving increases to $41\%$ when an additional trick is incorporated, in discretizing and re-scaling the classifier outputs and treating the resulting function as a hash function to a bit-map.

[2]By counting the minimal number of $A(S)$ sets required to cover all $S \subset U$.

$h_i : U \rightarrow \{1, \ldots, m\}$, $i = 1, \ldots, k$ such that $p(h_i(x) = j) = 1/m$ independent of prior hash values or input $x$. The Bloom Filter's memory $M \in [0, 1]^m$ is a binary string of length $m$ which is initialized to zero. Writes are performed by hashing an input $x$ to $k$ locations in $M$ and setting the corresponding bits to 1, $M[h_i(x)] \leftarrow 1$; $i = 1, \ldots, k$. For a given query $q$ the Bloom Filter returns true if all corresponding hashed locations are set to 1 and returns false otherwise: $Query(M, q) := M[h_1(q)] \wedge M[h_2(q)] \wedge \ldots \wedge M[h_k(q)]$. This incurs zero false negatives, as any previously observed input must have enabled the corresponding bits in $M$, however there can be false positives due to hash collisions. To achieve a false positive rate of $\epsilon$ with minimal space one can set $k = \log_2(1/\epsilon)$ and $m = n \log_2(1/\epsilon) \log_2 e$, where $e$ is Euler's number. The resulting space is a factor of $\log_2 e \approx 1.44$ from the optimal static lower bound given by Carter et al. (1978).

## 2.3 MEMORY-AUGMENTED NEURAL NETWORKS

Recurrent neural networks such as LSTMs retain a small amount of memory via the recurrent state. However this is usually tied to the number of trainable parameters in the model. There has been recent interest in augmenting neural networks with a larger external memory. The method for doing so, via a differentiable write and read interface, was first popularized by the Neural Turing Machine (NTM) (Graves et al., 2014) and its successor the Differentiable Neural Computer (DNC) (Graves et al., 2016) in the context of learning algorithms, and by Memory Networks (Sukhbaatar et al., 2015) in the context of question answering.

Memory Networks store embeddings of the input in separate rows of a memory matrix $M$. Reads are performed via a differentiable *content-based addressing* operation. Given a query embedding $q$ we take some similarity measure $D$ (e.g. cosine similarity, or negative euclidean distance) against each row in memory and apply a softmax to obtain a *soft* address vector $a \propto e^{D(q,M)}$. A read is then a weighted sum over memory $r \leftarrow a^T M$. The NTM and DNC use the same content-based read mechanism, but also learns to write. These models can arbitrate whether to write to slots in memory with similar content (content-based writes), temporally ordered locations, or unused memory.

When it comes to capacity, there has been consideration to scaling both the DNC and Memory Networks to very large sizes using sparse read and write operations (Rae et al., 2016; Chandar et al., 2016). However another way to increase the capacity is to increase the amount of compression which occurs in memory. Memory Nets can create compressive representations of each input, but cannot compress jointly over multiple inputs because they are hard-wired to write one slot per timestep. The NTM and DNC can compress over multiple slots in memory because they can arbitrate writes across multiple locations, but in practice seem to choose very sharp read and write addresses. The Kanerva Machine (Wu et al., 2018) tackles memory-wide compression using a distributed write scheme to jointly compose and compress its memory contents. The model uses content-based addressing over a separate learnable addressing matrix $A$, instead of the memory $M$, and thus learns *where* to write. We take inspiration from this scheme.

## 3 MODEL

One approach to learning set membership in one-shot would be to use a recurrent neural network, such as an LSTM or DNC. Here, the model sequentially ingests the $N$ elements to store, answers a set of queries using the final state, and is trained by BPTT. Whilst this is a general training approach, and the model may learn a compressive solution, it does not scale well to larger number of elements. Even when $N = 1000$, backpropagating over a sequence of this length induces computational and optimization challenges. For larger values this quickly becomes intractable. Alternatively one could store an embedding of each element $x_i \in S$ in a slot-based Memory Network. This is more scalable as it avoids BPTT, because the gradients of each input can be calculated in parallel. However Memory Networks are not a space efficient solution (as shown in in Section 5) because there is no joint compression of inputs.

We propose a memory model, the Neural Bloom Filter, that is both compressive and scalable. The network uses a purely *additive write* operation — i.e. no multiplicative gating or squashing — to update a real-valued memory matrix . An additive write operation can be seen as a continuous relaxation of the the Bloom Filter's logical OR write operation. The benefit of a purely additive write is that the resulting memory does not depend upon the ordering of the input (as addition is commutative) and the gradient with respect to each input can be computed in parallel.

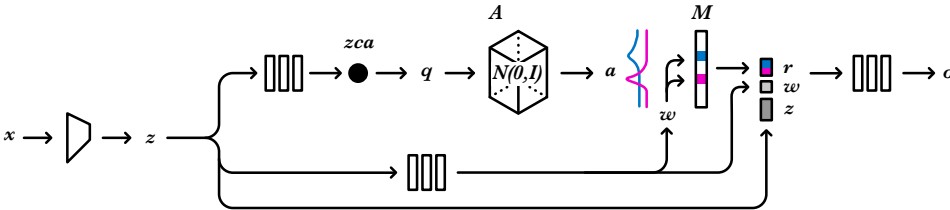

Figure 1: Overview of the Neural Bloom Filter architecture.

The network *addresses* memory by classifying which memory slots to read or write to via a softmax, conditioned purely on the input. We can think of this as a continuous relaxation of the Bloom Filter's hash function. To make the addressing more efficient for larger memory sizes, we sparsify the softmax by preserving only the top k components. When using a sparse address, we find the network can fixate on a subset of rows. This observation is common to prior sparse addressing work (Shazeer et al., 2017). We find sphering, often dubbed whitening, the addressing activations prior to the softmax (see Appendix F for an ablation) remedies this. Finally we make the final linear layer (denoted $A$) of the addressing softmax to be non-trainable. This is to ensure the number of trainable model parameters is independent of memory size. It also allows the memory to be resized at test-time.

The full architecture depicted in Figure 1 consists of a *controller network* which encodes the input to an embedding $z \leftarrow f_{enc}(x)$ and transforms this to a write word $w \leftarrow f_w(z)$ and a query $s \leftarrow f_q(z)$. An *addressing network* takes $s$ and performs a sphering transformation to produce a query $q$ with decorrelated dimensions. We used a moving average ZCA transformation. We choose ZCA over other decorrelation approaches as this worked well in practice, e.g. over PCA. We describe the exact moving zca transformation in full in Supplementary A.1. The address $a \leftarrow \sigma_k(q^T A)$ is computed over memory via content-based attention between $q$ and a non-learnable address matrix $A$. Here, $\sigma_k$ denotes a *sparse softmax* where the top $k$ similarities are retained. The addressing matrix $A \sim \mathcal{N}(\mathbf{0}, \mathbf{I})$ is chosen to be a fixed sample of Gaussian random variables. It simply represents a linear map from queries to memory addresses; we keep it non-trainable to avoid the coupling of model parameters and memory capacity.

A *write* is performed by running the controller and addressing network, and then additively writing $w$ to the top $k$ addresses in $a$, $M_{t+1} \leftarrow M_t + wa^T$. This scheme is inspired by the Bloom Filter, replacing a logical OR for an addition, and it ensures that we do not have to backpropagate-through-time (BPTT) over sequential writes. A *read* is performed by also running the controller and addressing network and retrieving the top $k$ addresses from $M$. These rows in memory are multiplied element-wise with the address weight $r \leftarrow M \odot a$ and are fed through an MLP with the residual inputs $w$ and $z$. We found this to be more powerful than the conventional read operation $r \leftarrow a^T M$ used by the DNC and Memory Networks, as it allows for non-linear interactions between rows in memory at the time of read. An overview of the operations is described below.

| **Controller network** | | **Write** | |
|---|---|---|---|
| $z \leftarrow f_{enc}(x)$ | Encoder network | (1) Run controller & address network | |
| $w \leftarrow f_w(z)$ | Write word | (2) $M_{t+1} \leftarrow M_t + wa^T$ | Additive write |
| $s \leftarrow f_q(z)$ | Raw query | **Read** | |
| **Addressing network** | | (1) Run controller & address network | |
| $q \leftarrow \texttt{moving\_zca}(s)$ | De-correlated query | (2) $r \leftarrow M \odot a$ | Read operation |
| $a \leftarrow \sigma_k(q^T A)$ | Sparse address | (3) $o \leftarrow f_{out}([r, w, z])$ | Output network |

To give an example network configuration, in our experiments we chose $f_{enc}$ to be a 3-layer CNN in the case of image input, and in the case of text input we choose $f_{enc}$ to be an LSTM with 128 hidden units. We chose $f_w$ and $f_q$ to be an MLP with a single hidden layer of size 128, and $f_{out}$ to be a 3-layer mlp with residual connections. We used leaky relus as the non-linearity.

We further discuss the motivation for decorrelation / sphering in the addressing network, and the model's relation to uniform hashing in Appendix A.2. We also discuss how the model could be implemented for $\mathcal{O}(\log m)$ time reads and writes, where $m$ is the size of memory, and with an $\mathcal{O}(1)$ network space overhead by avoiding the storage of $A$, the addressing matrix, in Appendix A.3.

## 4 SPACE COMPLEXITY

In this section we discuss space lower bounds for the approximate set membership problem when there is some structure to the storage or query set. This can help us formalize why and where neural networks may be able to beat classical lower bounds to this problem.

The $n \log_2 (1/\epsilon)$ lower bound from Carter et al. (1978) assumes that all subsets $S \subset U$ of size $n$, and all queries $q \in U$ have equal probability. Whilst it is instructive to bound this maximum-entropy scenario, which we can think of as 'worst case', most applications of approximate set membership e.g. web cache sharing, querying databases, or spell-checking, involve sets and queries that are not sampled uniformly. For example, the elements within a given set may be highly dependent, there may be a power-law distribution over queries, or the queries and sets themselves may not be sampled independently.

A more general space lower bound can be defined by an information theoretic argument from communication complexity (Yao, 1979). Namely, approximate set membership can be framed as a two-party communication problem between Alice, who observes the set $S$ and Bob, who observes a query $q$. They can agree on a shared policy $\Pi$ in which to communicate. For given inputs $S, q$ they can produce a transcript $A_{S,q} = \Pi(S,q) \in \mathcal{Z}$ which can be processed $g : \mathcal{Z} \to 0, 1$ such that $\mathbb{P}\left(g(A_{S,q}) = 1 | q \notin S\right) \leq \epsilon$. Bar-Yossef et al. (2004) shows that the maximum transcript size is greater than the mutual information between the inputs and transcript: $\max_{S,q} |A_{S,q}| \geq I(S, q; A_{S,q}) = H(S,q) - H(S, q | A_{S,q})$. Thus we note problems where we may be able to use less space than the classical lower bound are cases where the entropy $H(S, q)$ is small, e.g. our sets are highly non-uniform, or cases where $H(S, q | A_{S,q})$ is large, which signifies that many query and set pairs can be solved with the same transcript.

---

**Algorithm 1** Meta-Learning Training

---

1: **while** training iteration $i <$ budget **do**
2:     Sample task:
3:         Sample set to store $S \sim \mathcal{S}^{train}$
4:         Sample queries $x_1, x_2, \ldots, x_t \sim Q^{train}$
5:         Construct targets $y_1, y_2, \ldots y_t$ $s.t.$ $y_j = 1$ if $x_j \in S$ else 0
6:     Write entries to memory $M \leftarrow f_\theta^{write}(S)$
7:     Calculate query outputs $o_j = f_\theta^{read}(S, x_j), \ j = 1, \ldots, t$
8:     Calculate XE loss: $L = \sum_{j=1}^{t} y_j \log o_j + (1 - y_j)(1 - \log o_j)$
9:     Calculate $dL/d\theta$ (backpropagate through queries and writes)
10:     Update parameters $\theta_{i+1} \leftarrow \text{Optimizer}(\theta_i, dL/d\theta)$

---

## 5 EXPERIMENTS

Our experiments aim to explore scenarios inspired by real-world scenarios where there are varying levels of structure in the storage sets $S$ and queries $q$. We consider four neural architectures, the LSTM, DNC, Memory Network, and Neural Bloom Filter. An encoder architecture is shared across all models. For images we use a 3-layer convnet with a kernel size of 3 and 64 filters. For text we used a character LSTM with 128 hidden units.

The training setup follows the memory-augmented meta-learning training scheme of Vinyals et al. (2016), only here the task is familiarity classification versus image classification. The network samples tasks which involve classifying familiarity for a given storage set in one-shot. Meta-learning occurs as a two-speed process, where the model quickly learns to recognize a given storage set $S$ within a training episode via writing to a memory or state, and the model slowly learns to improve this fast-learning process by optimizing the model parameters $\theta$ over multiple tasks. We detail the training routine in Algorithm 1.

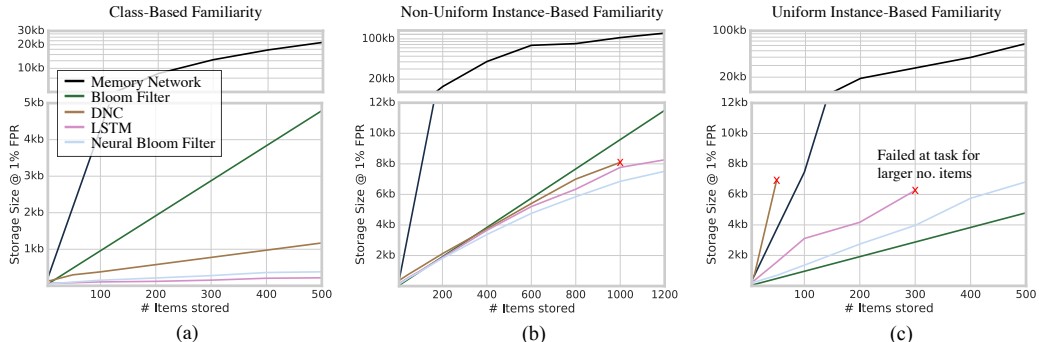

Figure 2: **Sampling strategies on MNIST.** Space consumption at 1% FPR.

For the RNN baselines (LSTM and DNC) the write operation corresponds to unrolling the network over the inputs and outputting the final state. For these models, the query network is simply an MLP classifier which receives the concatenated final state and query, and outputs a scalar logit. For the Memory Network the read and write operations are defined from the model, and for the Neural Bloom Filter the read and write operations are defined in Section 3.

## 5.1 SPACE COMPARISON

We compared the space (in bits) of the model's memory (or state) to a Bloom Filter at a given false positive rate. The false positive rate is measured empirically over a sample of queries for the learned models; for the Bloom Filter we employ the analytical false positive rate. Beating a Bloom Filter's space usage with the analytical false positive rate implies better performance for any given Bloom Filter library version (as actual Bloom Filter hash functions are not uniform), thus the comparison is fair. For each model we sweep over hyper-parameters relating to model size to obtain their smallest operating size at the desired false positive rate (for the full set, see Appendix D). Because the neural models can emit false negatives, we store these in a (ideally small) backup bloom filter, as proposed by Kraska et al. (2018); Mitzenmacher (2018a). We account for the space of this backup bloom filter, and add it to the space usage of the model's memory for parity (See Appendix B for further discussion). Thus the neural network must learn to output a small state in one-shot that can serve set membership queries at a given false positive rate, and emit a small enough number of false negatives such that the backup filter is also small — and the total size is considerably less than a Bloom Filter.

## 5.2 SAMPLING STRATEGIES ON MNIST

We chose three simple set membership tasks that have a graded level of structure and use images as an input type, namely images from MNIST. The input modality is not of principle importance, one could have designed similar tasks with textual input for example, but it is interesting to see how the model operate on images before moving to text. We experiment with three different levels of inherent structure to the sampling of sets and queries, however crucially all problems are approximate set membership tasks that can be solved by a Bloom Filter. They do not require generalization or familiarity to the sensitivity of a particular precision. *(1) Class-based familiarity*, each set of images is sampled with the constraint that they arise from the same randomly-sampled class. *(2) Non-uniform instance-based familiarity*, the images are sampled without replacement from an exponential distribution. *(3) Uniform instance-based familiarity*, where each subset contains images sampled uniformly without replacement. See Appendix E for further details on the task setup.

In the *class-based sampling* task we see in Figure 2a we see that the DNC, LSTM and Neural Bloom Filter are able to significantly outperform the classical Bloom Filter when images are sampled by class. The Memory Network is able to solve the task with a word size of only 2, however this corresponds to a far greater number of bits per element, 64 versus the Bloom Filter's 9.8 (to a total size of 4.8kb), and so the overall size was prohibitive. The DNC, LSTM, and Neural Bloom Filter are able to solve the task with 500 elements at 1.1kb , 217b, and 382b; a 4.3×, 22×, and 12× saving respectively. For the *non-uniform sampling task* in Figure 2b we see the Bloom Filter is preferable for less than 500 stored elements, but is overtaken thereafter. At 1000 elements the DNC,

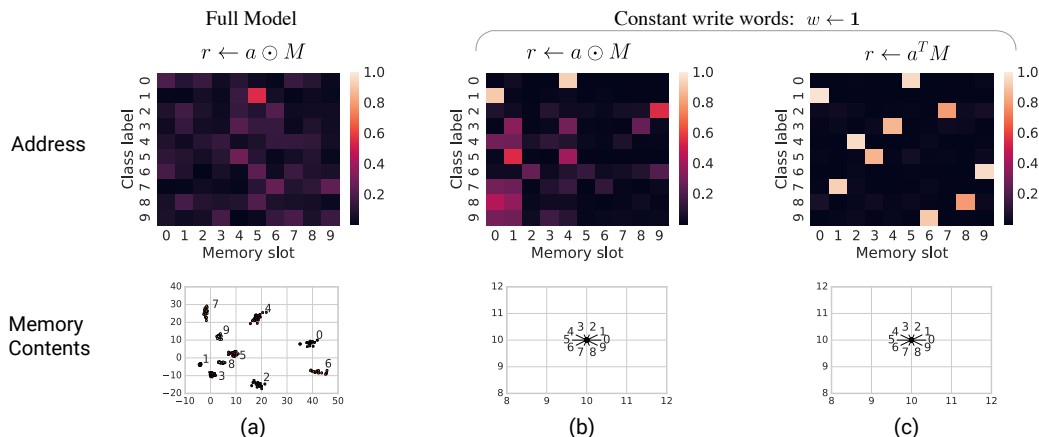

Figure 3: **Memory access analysis.** Three different learned solutions to class-based familiarity. We train three Neural Bloom Filter variants with 10 memory slots and visualize memory addressing $a$ and contents $\bar{M}$, broken down by class. Solutions share broad correspondence to known algorithms: (a) Bloom-g filters, (b) Bloom Filters, (c) Perfect hashing.

LSTM, and Neural Bloom Filter consume 7.9kb, 7.7kb, and 6.8kb respectively which corresponds to a 17.6%, 19.7%, and 28.6% reduction over the 9.6kb Bloom Filter. In the *uniform sampling task* shown in Figure 2c, there is no structure to the sampling of $S$. The two architectures which rely on BPTT essentially fail to solve the task at some threshold of storage size. The Neural Bloom Filter solves it with 6.8kb (using a memory size of 50 and word size of 2). If the Neural Bloom Filter were trained at low precision — say 5 bits per element which is a demonstrated capacity of RNN activations (Collins et al., 2016) — the overall space would be 5.8kb; still 20% larger than the Bloom Filter. The overall conclusion from these sets of experiments is that the classical Bloom Filter cannot be matched (or beaten) by a meta-learned memory-augmented neural network when there is no structure to the data, however in the case of imbalanced data, or highly dependent sets that share common attributes, we do see significant space savings.

## 5.3 MEMORY ACCESS ANALYSIS

We wanted to understand whether the Neural Bloom Filter is able to learn 'where' to store inputs in memory, as well as what to store. In other memory-augmented neural networks, such as the DNC and Memory Networks, there is no direct correspondence between the contents of an input and where it should be organized in memory. The Neural Bloom Filter has the ability do this, but it may not choose to in practice. We investigated this for the MNIST class-based familiarity task, giving the model 10 memory slots, each with a word size of 2 and a top k addressing of $k = 3$. We inspect three trained models; (1) the full model, (2) an ablated model where the write words are fixed to be a constant $w \leftarrow \mathbf{1}$, and (3) an ablated model with $w \leftarrow \mathbf{1}$ *and* a linear read operator $r \leftarrow a^T M$. We visualize the average write address and memory contents broken down by class. The full model, shown in Figure 3a learns to place some classes in particular slots, e.g. class $1 \rightarrow$ slot 5, however most are very distributed. Inspecting the memory contents it is clear the write word encodes a unique 2D token for each class. This solution bears resemblance with Bloom-g Filters (Qiao et al., 2011) where elements are spread across a smaller memory with the same hashing scheme as Bloom Filters, but a unique token is stored in each slot instead of a constant 1-bit value. With the model ablated to store only 1s in Figure 3b we see it chooses to allocate semantic addressing codes for some classes (e.g. 0 and 1) however for other classes it uses a distributed address. E.g. for class 3 the model prefers to uniformly spread its writes across memory slot 1, 4, and 8. The model solution is similar to that of Bloom Filters, with distributed addressing codes as a solution — but no information in the written words themselves. By inspecting the determinant of the average addresses, we find they are not linearly separable. Thus the output MLP $f_{out}$ computes a non-linearly separation. When we force the read operation to be linear in Figure 3c, this is no longer possible and the network learns to produce linearly separable addresses for each class. This solution has a correspondence with perfect hashing as now each (unobserved) input class will map to a unique slot in memory.

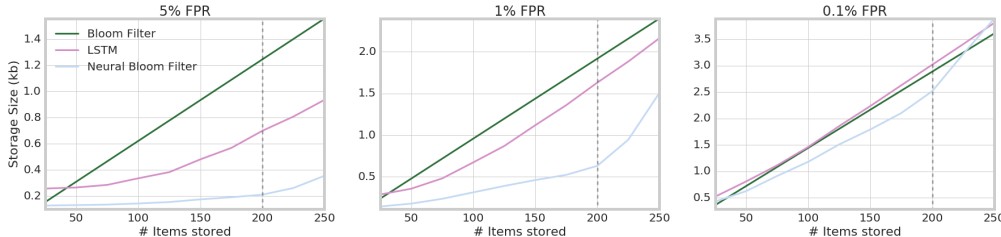

Figure 4: **Database task**. Models are trained up to sets of size 200 (dashed line).

## 5.4 DATABASE QUERIES

We look at a task inspired by database interactions. NoSQL databases, such as Bigtable and Cassandra, use a single string-valued row key, which is used to index the data. The database is comprised of a union of files (e.g. SSTables) storing contiguous row key chunks. Bloom Filters are used to determine whether a given query $q$ lies within the stored set. We emulate this setup by constructing a universe of strings, that is alphabetically ordered, and by sampling contiguous ranges (to represent a given SSTable). We choose unique tokens in GigaWord v5 news corpus to comprise as a universe, which consists of $2.5M$ words. We train models with up to $200$ unique strings in a set, and extrapolate to larger sizes. We evaluate the models on a held-out test set.

We see in Figure 4 that the neural architectures perform best for larger false positive rates. For a storage size of 200 we observe a $6\times$ space reduction ($200b$ vs $1.2kb$) when using the Neural Bloom Filter over the classical Bloom Filter. For smaller false positive rates, e.g. $0.1\%$ this gap reduces to a $20\%$ reduction (2.5kb vs 3kb) which decreases during the extrapolation phase. Interestingly the performance degradation of the LSTM is quite smooth between in-sample set sizes ($< 200$) and out-of-sample sizes ($> 200$). We hypothesize the LSTM is smoothly forgetting earlier inputs.

Finally, we train the Neural Bloom Filter with a storage size of $5000$ and compare it to Bloom Filters and Cuckoo Filters in Table 1, where we see $3 - 40\times$ space reduction. Here the LSTM was not able to learn the task; optimizing insertions via BPTT over a sequence of length 5000 did not result in a remotely usable solution. A storage size of 5000 is still small, but it is relevant to the NOSQL database-scenario where SSTables (say) are typically kept small. E.g. if the stored values were of size 100kB per row, we would expect 5000 unique keys in an average Bigtable SSTable. We furthermore considered the latency and throughput of such a learned model in this database setting. Both the query and insertion latencies of the Neural Bloom Filter are considerably higher than a classical Bloom Filter ($14ms$ vs $20ns$) however the maximum throughput of the Bloom Filter ($\approx 60K$ QPS) can be matched by the neural variant when run on a GPU (NVIDIA P6000). The same is not true of an LSTM, which peaks at a maximum insertion throughput of $4K$ even when the model is placed on the GPU — due to its sequential write scheme. See Appendix F.1 for more details on the speed benchmarks. The large database task validates the choice of simple write scheme for the Neural Bloom Filter. The additive write can be successfully optimized during training over larger storage sets as there is no BPTT, and it is an order of magnitude faster at evaluation time — even matching the throughput of a Bloom Filter.

## 6 RELATED WORK

There have been a large number of Bloom Filter variants published; from *Counting Bloom Filters* which support deletions (Fan et al., 2000), *Bloomier Filters* which store functions vs sets (Chazelle et al., 2004), *Compressed Bloom Filters* which use arithmetic encoding to compress the storage set (Mitzenmacher, 2002), and *Cuckoo Filters* which use cuckoo hashing to reduce redundancy within the storage vector (Fan et al., 2014). Although some of these variants focus on better compression, they do not achieve this by specializing to the data distribution.

One of the few works which do address data-dependence are *Weighted Bloom Filters* (Bruck et al., 2006; Wang et al., 2015). They work by modulating the number of hash functions used to store or query each input, dependent on its storage and query frequency. This requires estimating a large number of separate storage and query frequencies. The Neural Bloom Filter in contrast is a more

|  | Neural Bloom Filter | Bloom Filter | Cuckoo Filter |
|---|---|---|---|
| 5% FPR | 871b | 31.2kb | 33.1kb |
| 1% FPR | 1.5kb | 47.9kb | 45.3kb |
| 0.1% FPR | 24.5kb | 72.2kb | 62.6kb |

Table 1: **Database task**. Storing 5000 elements.

general solution to non-uniform $S$ sets. The encoder may be able to learn the statistics of the data distribution during the meta-learning phase and represent these in its parametric weights. However it can also take advantage of dependent sets where the entropy $H(S)$ may be low but the frequency of each element is uniform (e.g. such as row keys in a database).

Sterne (2012) proposes a neurally-inspired set membership data-structure that works by replacing the randomized hash functions with a randomly-wired computation graph of *OR* and *AND* gates. The false positive rate is controlled analytically by modulating the number of gates and the overall memory size. However there is no learning or specialization to the data with this setup. Bogacz & Brown (2003) investigates a learnable neural familiarity module, which serves as a biologically plausible model of familiarity mechanisms in the brain, namely within the perirhinal cortex. However this has not shown to be empirically effective at exact matching.

Kraska et al. (2018) consider the use of a neural network to classify the membership of queries to a fixed set $S$. Here the network itself is more akin to a perfect hashing setup where multiple epochs are required to find a succinct holistic representation of the set, which is embedded into the weights of the network. In their case this search is performed by gradient-based optimization. We emulate their experimental comparison approach but instead propose a memory architecture that represents the set as activations in memory, versus weights in a network.

Mitzenmacher (2018a) discusses the benefits and draw-backs of a learned bloom filter; distinguishing the empirical false positive rate over the distribution of sets $S$ versus the conditional false positive rate of the model given a particular set $S$. In this paper we focus on the empirical false positive rate because we wish to exploit redundancy in the data and query distribution. Mitzenmacher (2018b) considers a different way to combine classical and learned bloom filters; by 'sandwiching' the learned model with a pre-filter classical bloom filter and a post-filter classical bloom filter. This is seen to be more efficient for learned models with a larger false positive rate.

## 7 CONCLUSION

In many situations neural networks are not a suitable replacement to Bloom Filters and their variants. The Bloom Filter is robust to changes in data distribution, and adversarial attacks, because it delivers a bounded false positive rate for any sampled subset, unlike a neural network. However in this paper we consider the questions, "When might a neural network provide better compression than a Bloom Filter?" and "What kind of neural architecture is practical?". We see that a model which uses an external memory with an adaptable capacity, avoids BPTT with a feed-forward write scheme, and learns to address its memory, is the most promising option in contrast to popular memory models such as DNCs and LSTMs. We term this model the Neural Bloom Filter due to the analogous incorporation of a hashing scheme, commutative write scheme, and multiplicative read mechanism.

The Neural Bloom Filter relies on settings where cost of learning to query is possible and will be a net benefit to a population of existing bloom filters. That is, because we rely on meta-learning, we need situations where we have an off-line dataset (both of stored elements and queries) that is similar enough to future data that we wish to store. In the case of a large database we think this is warranted, a database with $100,000$ separate set membership data structures will benefit from a single (or periodic) meta-learning training routine that can run on a single machine and sample from the currently stored data, generating a large number of efficient data-structures. We envisage the space cost of the network to be amortized by sharing it across many neural bloom filters, and the time-cost of executing the network to be offset by the continuous acceleration of dense linear algebra on modern hardware, and the ability to batch writes and queries efficiently. A promising future direction would be to investigate the feasibility of this approach in a production system.

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

## A    FURTHER MODEL DETAILS

### A.1    MOVING ZCA

The moving ZCA was computed by taking moving averages of the first and second moment, calculating the ZCA matrix and updating a moving average projection matrix $\theta_{zca}$. This is only done during training, at evaluation time $\theta_{zca}$ is fixed. We describe the update below for completeness.

$$\text{Input: } s \leftarrow f_q(z) \tag{1}$$
$$\mu_{t+1} \leftarrow \gamma\mu_t + (1-\gamma)\bar{s} \qquad \text{First moment} \tag{2}$$
$$\Sigma_{t+1} \leftarrow \gamma\Sigma_t + (1-\gamma)\, s^T s \qquad \text{Second moment} \tag{3}$$
$$U, s, {}_{-} \leftarrow \texttt{svd}(\Sigma - \mu^2) \qquad \text{Calculate singular values} \tag{4}$$
$$W \leftarrow UU^T/\sqrt{(s)} \qquad \text{Calculate full ZCA matrix} \tag{5}$$
$$\theta_{zca} \leftarrow \eta\theta_{zca} + (1-\eta)W \qquad \text{ZCA moving average} \tag{6}$$
$$q \leftarrow s\,\theta_{zca} \tag{7}$$

In practice we do not compute the singular value decomposition at each time step to save computational resources, but instead calculate it and update $\theta$ every $T$ steps. We scale the discount in this case $\eta' = \eta/T$.

### A.2    RELATION TO UNIFORM HASHING

We can think of the decorrelation of $s$, along with the sparse content-based attention with $A$, as a hash function that maps $s$ to several indices in $M$. For moderate dimension sizes of $s$ (256, say) we note that the Gaussian samples in $A$ lie close to the surface of a sphere, uniformly scattered across it. If $q$, the decorrelated query, were to be Gaussian then the marginal distribution of nearest neighbours rows in $A$ will be uniform. If we chose the number of nearest neighbours $k = 1$ then this implies the slots in $M$ are selected independently with uniform probability. This is the exact hash function specification that Bloom Filters assume. Instead we use a continuous (as we choose $k > 1$) approximation (as we decorrelate $s \to q$ vs Gaussianize) to this uniform hashing scheme, so it is differentiable and the network can learn to shape query representations.

### A.3    EFFICIENT ADDRESSING

We discuss some implementation tricks that could be employed for a production system. One can avoid the linear-time addressing operation $\sigma_k(q^T A)$ by using an approximate k-nearest neighbour index, such as locality-sensitive hashing, to extract the nearest neighbours in $A$. The use of an approximate nearest neighbour index has been empirically considered for scaling memory-augmented neural networks (Rae et al., 2016; Kaiser et al., 2017) however this was used for attention on $M$ directly. As $M$ is dynamic the knn requires frequent re-building as memories are stored or modified. This architecture is simpler — $A$ is fixed and so the approximate knn can be built once.

To ensure the size of the network (which can be shared across many memory instantiations) is independent of the number of slots in memory $m$ we can avoid storing $A$. Because it is a fixed sample of random variables that are generated from a deterministic random number generator we can instead store a set of integer seeds that can be used to re-generate the rows of $A$. We can let the $i$-th seed $c_i$, say represented as a 16-bit integer, correspond to the set of 16 rows with indices $16i, 16i + 1, \ldots, 16i + 15$. If these rows need to be accessed, they can be regenerated on-the-fly by $c_i$. The total memory cost of $A$ is thus $m$ bits, where $m$ is the number of memory slots[3].

Putting these two together it is possible to query and write to a Neural Bloom Filter with $m$ memory slots in $\mathcal{O}(\log m)$ time, where the network consumes $\mathcal{O}(1)$ space. It is worth noting, however, the Neural Bloom Filter's memory is often much smaller than the corresponding classical Bloom Filter's memory, and in many of our experiments is even smaller than the number of unique elements to store. Thus dense matrix multiplication can still be preferable - especially due to its acceleration on GPUs and TPUs (Jouppi et al., 2017) - and a dense representation of $A$ is not inhibitory. As model

---

[3]One can replace 16 with 32 if there are more than one million slots

optimization can become application-specific, we do not focus on these implementation details and use the model in its simplest setting with dense matrix operations.

## B    SPACE COMPARISON

For each task we compare the model's memory size, in bits, at a given false positive rate — usually chosen to be 1%. For our neural networks which output a probability $p = f(x)$ one could select an operating point $\tau_\epsilon$ such that the false positive rate is $\epsilon$. In all of our experiments the neural network outputs a memory (state) $s$ which characterizes the storage set. Let us say `SPACE(f, ` $\epsilon$`)` is the minimum size of $s$, in bits, for the network to achieve an average false positive rate of $\epsilon$. We could compare `SPACE(f,`$\epsilon$`)` with `SPACE(Bloom Filter,`$\epsilon$`)` directly, but this would not be a fair comparison as our network $f$ can emit false negatives.

To remedy this, we employ the same scheme as Kraska et al. (2018) where we use a 'backup' Bloom Filter with false positive rate $\delta$ to store all false negatives. When $f(x) < \tau_\epsilon$ we query the backup Bloom Filter. Because the overall false positive rate is $\epsilon + (1 - \epsilon)\delta$, to achieve a false positive rate of at most $\alpha$ (say 1%) we can set $\epsilon = \delta = \alpha/2$. The number of elements stored in the backup bloom filter is equal to the number of false negatives, denoted $n_{fn}$. Thus the total space can be calculated, `TOTAL_SPACE(f,`$\alpha$`)` `=` `SPACE(f,`$\frac{\alpha}{2}$`)` `+` $n_{fn}$ `*` `SPACE(Bloom Filter,`$\frac{\alpha}{2}$`)`. We compare this quantity for different storage set sizes.

## C    MODEL SIZE

For the MNIST experiments we used a 3-layer convolutional neural network with 64 filters followed by a two-layer feed-forward network with $64\&128$ hidden-layers respectively. The number of trainable parameters in the Neural Bloom Filter (including the encoder) is $243,437$ which amounts to 7.8Mb at 32-bit precision. We did not optimize the encoder architecture to be lean, as we consider it part of the library in a sense. For example, we do not count the size of the hashing library that an implemented Bloom Filter relies on, which may have a chain of dependencies, or the package size of TensorFlow used for our experiments. Nevertheless we can reason that when the Neural Bloom Filter is 4kb smaller than the classical, such as for the non-uniform instance-based familiarity in Figure 2b, we would expect to see a net gain if we have a collection of at least $1,950$ data-structures. We imagine this could be optimized quite significantly, by using 16-bit precision and perhaps using more convolution layers or smaller feed-forward linear operations.

For the database experiments we used an LSTM character encoder with 256 hidden units followed by another 256 feed-forward layer. The number of trainable parameters in the Neural Bloom Filter $419,339$ which amounts to 13Mb. One could imagine optimizing this by switching to a GRU or investigating temporal convolutions as encoders.

## D    HYPER-PARAMETERS

We swept over the following hyper-parameters, over the range of memory sizes displayed for each task. We computed the best model parameters by selecting those which resulted in a model consuming the least space as defined in Appendix B. This depends on model performance as well as state size.

The Memory Networks memory size was fixed to equal the input size (as the model does not arbitrate what inputs to avoid writing).

| | |
|---|---|
| Memory Size (DNC, NBF) | $\{2, 4, 8, 16, 32, 64, 96, 128\}$ |
| Word Size (MemNets, DNC, NBF) | $\{2, 4, 6, 8, 10\}$ |
| Hidden Size (LSTM) | $\{2, 4, 8, 16, 32, 64, 96, 128\}$ |
| Sphering Decay $\eta$ (NBF) | $\{0.9, 0.95, 0.99\}$ |
| Learning Rate (all) | $\{$1e-4, 5e-5$\}$ |

Table 2: Hyper-parameters considered

## E    EXPERIMENT DETAILS

For the class-based familiarity task, and uniform sampling task, the model was trained on the training set and evaluated on the test set. For the class-based task sampling, a class is sampled at random and $S$ is formed from a random subset of images from that class. The queries $q$ are chosen uniformly from either $S$ or from images of a different class.

For the non-uniform instance-based familiarity task we sampled images from an exponential distribution. Specifically we used a fix permutation of the training images, and from that ordering chose $p(i_{th}$ image$) \propto 0.999^i$ for the images to store. The query images were selected uniformly. We used a fixed permutation (or shuffle) of the images to ensure most probability mass was not placed on images of a certain class. I.e. by the natural ordering of the dataset we would have otherwise almost always sampled $0$ images. This would be confounding task non-uniformity for other latent structure to the sets. Because the network needed to relate the image to its frequency of occurence for task, the models were evaluated on the training set. This is reasonable as we are not wishing for the model to visually generalize to unseen elements in the setting of this exact-familiarity task. We specifically want the network weights to compress a map of image to probability of storage.

For the database task a universe of $2.5M$ unique tokens were extracted from GigaWord v5. We shuffled the tokens and placed 2.3M in a training set and $250$K in a test set. These sets were then sorted alphabetically. A random subset, representing an SSTable, was sampled by choosing a random start index and selecting the next $n$ elements, which form our set $S$. Queries are sampled uniformly at random from the universe set. Models are trained on the training set and evaluated on the test set.

## F    EFFECT OF SPHERING

We see the benefit of sphering in Figure 5 where the converged validation performance ends up at a higher state. Investigating the memory fill rate, we see this is because the model is ignoring many more rows than is necessary. This is likely due to the network fixating on rows it has accessed with sparse addressing, and ignoring rows it has otherwise never touched — a phenomena noted in Shazeer et al. (2017). The model finds a local minima in continually storing and accessing the same rows in memory. The effect of sphering is that the query now appears to be Gaussian (up to the first two moments) and so the nearest neighbour in the address matrix A (which is initialized to Gaussian random variables) will be close to uniform. This results in a more uniform memory access (as seen in Figure 5b) which significantly aids performance (as seen in Figure 5a).

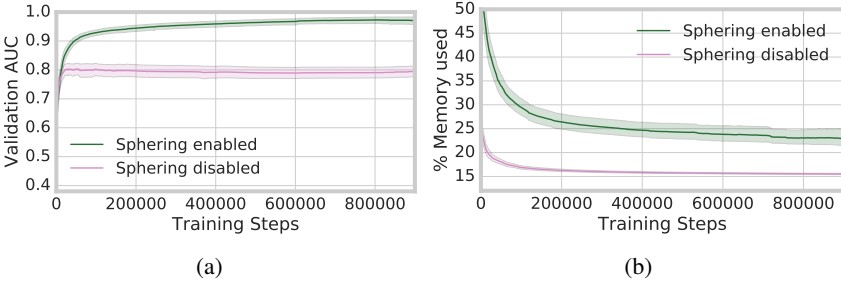

(a)                                                      (b)

Figure 5: The effect of sphering the query vector on performance in the uniform MNIST sampling task.

### F.1    TIMING BENCHMARK

We provide a brief comparison of run-time (in terms of latency and max-throughput) between a classical Bloom Filter and a Neural Bloom Filter. In several applications of Bloom Filters, the actual query latency is not in the critical path of computation. For example, for a distributed database, the network latency and disk access latency for one tablet can be orders of magnitude greater than the in-memory latency of a Bloom Filter query. For this reason, we have not made run-time a point of focus in this study, and it is implicitly assumed that the neural network is trading off greater latency for less space. However it is worth checking whether run-time could be prohibitive.

| | Query + Insert Latency | | Query Throughput (QPS) | | Insert Throughput (QPS) | |
|---|---|---|---|---|---|---|
| | CPU | GPU | CPU | GPU | CPU | GPU |
| Bloom Filter* | 0.02ms | - | 61K | - | 61K | - |
| NBF | 5.1ms | 13ms | 3.5K | 58.5K | 3.2K | 58K |
| LSTM | 5.0ms | 13ms | 3.1K | 58.7k | 2.4K | 4.6K |

Table 3: Latency for a single query, and throughput for a batch of 10,000 queries. *Bloom Filter benchmark taken from 'query-efficient' Bloom Filter (Chen et al., 2007).

We use the Neural Bloom Filter network architecture for the large database task (Table 1). The network uses an encoder LSTM with $256$ hidden units over the characters, and feeds this through a $256$ fully connected layer to encode the input. A two-layer 256-hidden-unit MLP is used as the query architecture. The memory and word size is $8$ and $4$ respectively, and so the majority of the compute is spent in the encoder and query network. We compare this with an LSTM containing $32$ hidden units. We benchmark the single-query latency of the network alongside the throughput of a batch of queries, and a batch of inserts. The Neural Bloom Filter and LSTM is implemented in TensorFlow without any custom kernels or specialized code. We benchmark it on the CPU (Intel(R) Xeon(R) CPU E5-1650 v2 @ 3.50GHz) and a GPU (NVIDIA Quadro P6000). We compare to empirical timing results published in a query-optimized Bloom Filter variant (Chen et al., 2007).

We see in Table 3. that the query and insert latency of the Neural Bloom Filter sits at $5$ms on the CPU, around $400\times$ slower than the classical Bloom Filter. The LSTM requires roughly the same latency. However when multiple queries are received, the operations batch well and the model is only around $20\times$ slower for both the Neural Bloom Filter and LSTM — as these can be batched in parallel on the CPU. If we are able to use a GPU, the throughput further increases to roughly parity with the classical Bloom Filter. This is a very rough comparison, however it leads to the conclusion that a Neural Bloom Filter could be deployed in scenarios with high query load without a catastrophic decrease in throughput.

For insertions we see the same insertion throughput of $\approx 58K$ is maintained for the Neural Bloom Filter on the GPU as the model uses batchable insertion operations that can be parallelized. The use of a purely additive write was primarily chosen to avoid optimizing the model with backpropagation-through-time, however here we see it is beneficial for computational efficiency at evaluation time. However here we see a large difference between the NBF and the LSTM, which uses a strictly-sequential write scheme. Even with a powerful NVIDIA P6000 GPU the LSTM's maximum insertion throughput is $4.6K$, over an order of magnitude lower. Thus we conclude that even if we were able to train an LSTM to perform exact set membership on the larger database task to a good space performance — which we were not able to do due to the difficulty of training an RNN over rich sequences of length $5000$ — it would serve insertions at over an order of magnitude slower than a Bloom Filter, even with accelerated hardware.

