# OpenReview forum: "Meta-Learning Neural Bloom Filters"
_ICLR.cc/2019/Conference_

### Official Review · AnonReviewer2 · 2018-10-30
**Details of the architecture not well motivated**

**Rating:** 7
**Confidence:** 3

**Review:**

The paper proposes a learnable bloom filter architecture. While the details of the architecture seemed a bit too complicated for me to grasp (see more on this later), via experiments the authors show that the learned bloom filters are more compact that regular bloom filters and can outperform other neural architectures when it comes to retrieving seen items.

A bloom filter is fairly simple, K hash functions hash seen items into K bit vectors. During retrieval, if all of the bits hashed to are 1 then we say we've seen the query. I think there's simpler ways to derive a continuous, differentiable version of this which begs the question why the authors chose a relatively more elaborate architecture involving ZCA transform and first/second moments. Perhaps the authors need to motivate their architecture a bit better.

In their experiments, a simple LSTM seems to perform remarkably well (it is close to the best in 2 (a), (b); and crashes in (c) but the proposed technique is also outperformed by vanilla bloom filters in (c)). This is not surprising to me since LSTMs are remarkably good at remembering patterns. Perhaps the authors would like to comment on why they did not develop the LSTM further to remedy it of its shortcomings. Some of the positive results attained using neural bloom filters is a bit tempered by the fact that the experiments were using a back up bloom filter. Also, the neural bloom filters do well only when there is some sort of querying pattern. All of these details would seem to reduce the applicability of the proposed approach.

The authors have addressed most (if not all) of my comments in their revised version. I applaud the authors for being particularly responsive. Their explanations and additional experiments go a long way towards lending the insights that were missing from the original draft of the paper. I have upped my rating to a 7.

---

> ### Author Response · Authors · 2018-11-17
> **Model motivation and applicability.**
>
> Thank you for your review, and for your keen eye to detail.
>
> Re. “why not develop further an LSTM”. We are principally interested in whether it is possible to learn a compressive set membership data-structure in one-shot. Because many applications of Bloom Filters are in highly dynamic settings (e.g. databases), the requirement that a network may be able to beat a Bloom Filter with only a single computational pass over the data is quite important. It wasn’t clear from the beginning of this research project (to us and our peers) whether it would be possible, and if so - in what setting. Thus we feel that it would be a worthwhile scientific contribution to show this is the case with any model --- even an LSTM.
>
> Firstly, it is worth noting the LSTM is non-trivially less efficient for the database task even when the sequence length is quite short. But the real issue with an LSTM and other RNNs (partially covered in the Reviewer 3 response) is that it they are difficult to scale to larger set sizes, because one has to BPTT over the entire input sequence linearly (elements of our storage size S) during training. Say S contains 5,000 elements… One would have to train with sequences of length 5,000, insert all elements sequentially and BPTT over the 5,000 long sequence. Training is way too slow, and the optimization problem becomes intractable (network fails to learn). Furthermore the LSTM has quadratic computation cost with respect to the hidden state size. Since set membership is order-invariant, it seemed preferable to try out a memory architecture which does not rely on sequential computation and BPTT (like a memory network) that is still very compressive (unlike a memory network).
>
> Our mistake in the original exposition, which you rightly point out, is that we have presented our solution (the architecture) without much of the motivation that lead to its incarnation. We have re-written the model section to remedy this. But we will also briefly state the model motivation here:
>
> - We want a simple write scheme and no BPTT -> additive write. (it’s order-invariant, and alike to the Bloom Filter’s logical-or write).
>
> - We want the network to choose where to write, as well as what to write -> address is a softmax over memory based on content. (alike to the Kanerva Machine Wu et al. (2018))
>
> - We want the network’s network trainable parameters to be small and independent of memory size -> make the addressing matrix A non-trainable.
>
> - We want the addressing to be efficient -> make it sparse (alike to Rae et al. (2016)).
>
> - We found a sparse address led to the network fixating on a subset of memory -> whiten the query vector.
>
> Whitening (or sphering) may appear complex but was only necessary if one adopts the sparse attention for efficiency. We implemented it in four lines of TensorFlow code, so at least it is not too complex from an engineering standpoint. Whitening has been used within deep learning literature before, e.g. “natural neural networks” [1] . An alternative to whitening would be to use a “flow” such as real NVP [2] which actually transforms the query to something which appears to be truly gaussian. Crucially, this was a trick to get sparse attention working, if one wishes to avoid sparse attention and just use the full softmax over memory then this side-detail of whitening can be ignored.
>
> -- Re. “Also, the neural bloom filters do well only when there is some sort of querying pattern. All of these details would seem to reduce the applicability of the proposed approach.”
>
> Fortunately the proposed approach does well if there is structure to the query pattern *or* storage set. In the case of the database task, our queries are picked uniformly from the universe --- there is not much structure. However there is structure to the storage sets (which represent row keys in an disk file within a database) and this is why our approach outperforms the classical data-structures so significantly.
>
> More generally we think the research area of using neural networks to replace data-structures, in this case a bloom filter, is so exciting because (we would argue) they are very rarely applied to data that contains no structure. Using a neural network to exploit redundancy and save space feels like a very impactful thing to do, and thought leaders within Computer Science (e.g. Jeff Dean, a co-author of the kraska et al. 2018 paper) appear to believe so. There are patterns to the rowkey schema that is used within our databases, there are patterns to blacklist URLs and IPs within our firewalls, there are patterns to our search queries.
>
> We have re-written the model and experiment section to address your concerns!
>
> [1] https://deepmind.com/research/publications/natural-neural-networks/
> [2] https://arxiv.org/abs/1605.08803

---

> > ### Author Response · Authors · 2018-11-20
> > **Backup Bloom Filter**
> >
> > Sorry we realize we did not address this comment,
> >
> > "Some of the positive results attained using neural bloom filters is a bit tempered by the fact that the experiments were using a back up bloom filter."
> >
> > Actually when comparing the space of our one-shot model versus a Bloom Filter; we compute the size of the state (in bits) *plus* the size of the backup Bloom Filter which stores the false negatives. The backup Bloom Filter (which *only* stores false negatives) thus must be very small in comparison with the original Bloom Filter for the total space of the neural bloom filter to be smaller. In the case of the database task where we see >30x space reduction, it is clearly negligible.
> >
> > We only use a backup filter to ensure an apples-to-apples comparison between the neural bloom filter and bloom filter (i.e. a guaranteed 0% false negative rate). For applications where a small false negative rate is acceptable, one could avoid using the backup bloom filter completely. We have clarified this in the text in the experiments section. In terms of speed, the backup bloom filter does not add latency to queries because it can be queried in parallel to the neural bloom filter.

---

### Official Review · AnonReviewer3 · 2018-11-01
**Interesting topic, some concerns**

**Rating:** 6
**Confidence:** 4

**Review:**

SUMMARY
The paper proposes a neural network based architecture to solve the approximate set membership problem, in the distributional setting where the in-set and out-of-set elements come from two unknown and possibly different distributions.


COMMENTARY
The topic of the paper is interesting, and falls into the popular trend of enhancing classical data structures with learning algorithms. For the approximate set membership problem, this approach was already suggested by (Kraska et al. 2018) and studied further in (Mitzenmacher 2018a,b). The difference in the current paper is that the proposed approach relies on "meta-learning", apparently to facilitate online training and/or learning across multiple sets arising from the same distribution; this is what I gather from the introduction, even though as I write below, I feel this point is not properly explained.

My main issue with the paper is that its conceptual contribution seems limited and unclear. It suggests a specific architecture whose details seem mostly arbitrary, or at least this is the impression the reader is left with, as the paper does rather little in terms of discussing and motivating them or putting them in context. Moreover, since the solution ultimately relies on a backup Bloom Filter as in (Kraska et al. 2018), it is hard to not view it as just an instantiation of the model in (Kraska et al. 2018, Mitzenmacher 2018a) with a different plugging of learning component. It would help to flesh out and highlight what the authors claim are the main insights of the paper.

Another issue I suggest revising pertains to the writing. The problem setting is only loosely sketched but not properly defined. How exactly do different subsets coming into play? Specifically, the term "meta-learning" appears in the title and throughout the paper, but is never defined or explained. The authors should write out what exactly they mean by this notion and what role it plays in the paper. This is important since to my understanding, this is the main point of departure from the aforementioned recent works on learning-enhanced Bloom Filters.

The experiments do not seem to make a strong case for the empirical advantage of the Neural Bloom Filter. They show little to no improvement on the MNIST tasks, and some improvement on a non-standard database related task. One interesting thing to look at would be the workload partition between the learning component and the backup filter, meaning what is the rate of false negatives emitted by the former and caught by the latter, and how the space usage breaks down between them (vis-a-vis the formula in Appendix B). For example, it seems plausible that on the class familiarity task, the learning component simply learns to be a binary classifier for the chosen two MNIST classes and mostly ignores the backup filter, whereas in the uniform distribution setting, the learning component only memorizes a small number of true and false positives and defers almost the entire task to the backup filter. I am not sure what to expect on the intermediate exponential distribution task.

Other comments/questions:
1. For the classical Bloom Filter, do the results reported in the experimental plots reflect the empirical false-positive rate measured in the experiment, or just the analytic bound?
2. On that note, it is worth noting that the false positive rate of the classical Bloom Filter is different than the one you report for the neural-net based architectures. The Bloom Filter FP probability is over its internal randomness (i.e. its hash functions) and is independent of the distribution of queries, which need not be randomized at all. For the neural-net based architectures, the measured FP rate is w.r.t. a specific distribution of queries. See the discussion in (Mitzenmacher 2018a), sections B-C.
3. The works (Mitzenmacher 2018a,b) should probably at least be referenced in the related work section.


CONCLUSION
While I like the overall topic of the paper, I currently find the conceptual contribution to be too thin, raising doubts on novelty and significance. In addition, the presentation is somewhat lacking in clarity, and the practical merit is not well established. Notwithstanding the public nature of ICLR submissions, I would suggest more work on the paper prior to publication.


REFERENCES
M. Mitzenmacher, A Model for Learned Bloom Filters and Related Structures, 2018, see https://arxiv.org/pdf/1802.00884.pdf.
M. Mitzenmacher, Optimizing Learned Bloom Filters by Sandwiching, 2018, see https://arxiv.org/pdf/1803.01474.pdf.

(Update: score revised, see below.)

---

> ### Author Response · Authors · 2018-11-17
> **R3 response**
>
> Thank you for this thoughtful and comprehensive review.
>
> -- We agree the recent Mitzenmacher arxiv posts should have been included in the related works, and they have now been added.
>
> -- Re. ‘strong empirical case for NBF…’ The fact that an LSTM does well on the MNIST class-based familiarity task is a useful data-point. However we do see a substantial gain for the database task. However the main problem with RNNs such as the LSTM (and DNC) is that they are not scalable. need to be trained to store N items by ingesting the N elements sequentially, and then backpropagating over the entire sequence. For large N this does not end up being scalable; e.g. for the large database task (Table 1) where N = 5,000. Thus we develop a memory model that does not rely on BPTT (alike to memory networks) but is compressive (unlike memory networks).
>
> The crucial design-point of the model is that it uses a commutative write operation (addition) which is much simpler than the DNC & LSTM write (e.g. no gating, no squashing of the state) and is like a continuous relaxation of the Bloom Filter’s write (logical or). A simple additive write scheme also means the model will produce the same external memory M regardless of the ordering of the inputs (because addition is commutative) which makes sense given that familiarity does not depend upon input ordering, thus we also do not get strange effects where older inputs have much worse performance than newer inputs (which will occur with an RNN).  We discuss the model’s motivation more explicitly in the revised text.
>
> -- Re. “One interesting thing to look at would be the workload partition between the learning component and the backup filter”. This is a very interesting question you ask here. Your intuition is absolutely pretty well for class-based familiarity, the backup filter is used where the encoder essentially miss-classifies a character (so it is very lightly used). For uniform sampling, the model essentially captures a small random subset of inputs but mostly relies on the backup bloom filter. For the imbalanced data the model appears to store and characterise well the ‘heavy hitter’ i.e. frequent elements in the state memory and uses the backup bloom filter for infrequent elements.
>
> -- Re. ‘problem setting is loosely sketched…’ - the reviewer is correct, we originally wrote the paper for readers familiar with the recent one-shot memory-augmented meta-learning literature (e.g. matching networks [vinyals et al. 2016], MANN [santoro et al. 2016]) but unfamiliar with Bloom Filters. This was an unfortunate choice, we have thus expanded on what we mean by meta-learning and described how the training regime works. It is the exact same training regime as that in vinyals et al. 2016 and many follow-on works, only the classification problem is set membership, versus image classification. We have added a subsection with further explanation and an algorithm box with a succinct summary of the meta-learning training setup.
>
> We will  just briefly summarize the training setup here. We have a collection of sets {S_1, S_2, , … S_m} reserved for training (each set contains n points to insert); and a collection of queries Q = {q1, q2, …, qL} and targets yi = 1 if qi in S and 0 otherwise. In the example of a database we can think of a given set Si = {k1, …, kN} as a set of rowkeys for a given file on disk (e.g. SSTable). We have many sets because we have many files; for training we have reserved some for an offline training routine.
>
> During training we calculate M = f_write(S), and then we calculate oi = f_read(S, qi), our query responses having observed the set S only once. We calculate the cross-entropy loss L = \sumi yi log(oi) + (1 - yi)log(1-oi) and backprogate through the network (through the parameters controlling both the read, write, and encoder networks). One can consider the creation of M = f_write(S) as a fast one-shot learning procedure; the network learns a state which can help it solve the classification problem, “is q in S?” in one-shot. The slow-moving ‘meta-learning’ process is in the network parameters, which are slowly being optimized over several set membership tasks, i.e. several different sets S_1:m, to be effective at one-shot classification. At test time, when we observe a new subset (or stream of elements) we can insert them with f_write in one-shot and the resulting data-structure is the external memory, M.
>
> -- Re. Bloom Filter space usage, we indeed used the analytical bound. We have clarified this in the text. We feel this is fair as it makes the task of beating a Bloom Filter’s space performance slightly more difficult (as the analytic bound is slightly more compressive than in-practice), and it absolves any dispute over the choice of Bloom Filter library / choice of hash function etc.
>
> -- We have clarified the false positive rate is with respect to the distribution of queries in the text.

---

> > ### Comment · AnonReviewer3 · 2018-11-24
> > **Response to revision**
> >
> > I thank the authors for their detailed responses and revision.
> > - The revision to section 5 explaining the training procedure is helpful.
> > - The revision to section 3 is also helpful. It may have helped to go even further in explaining the objectives of the memory access learning task (as in the response to Rev2) with analogs to BF, and distinguishing them from some aspects that seem like engineering details, as the former is (to me) the conceptually significant portion of the paper.
> > - I still could not find where it is stated that the BF plots are analytic; my apology if I missed it. This is not a major issue, and I understand the choice to use the theoretical bound, but there is some discord in including an analytic curve next to empirical curves on the same plot without clearly marking it as such, as it may give a wrong impression as to what the reader is seeing (not an actual experiment, but an estimate of what an experiment would have yielded based on probabilistic concentration).
> > - I have revised my score to 6.

---

> > > ### Author Response · Authors · 2018-11-24
> > > **Re. Re.**
> > >
> > > Thanks R3 for reading the revision and rebuttal.
> > >
> > > Small point re. analytical bounds --- we have these two sentences in S 5.1 which was perhaps not pushed into the revision (although I see it now) however we could also put (analytical) in the plot legends if you think this is worthwhile.
> > >
> > > "The false positive rate is measured empirically over a sample of queries for the learned models; for the Bloom Filter we employ the analytical false positive rate. Beating a Bloom Filter’ss pace usage with the analytical false positive rate implies better performance for any given BloomFilter library version (as actual Bloom Filter hash functions are not uniform), thus the comparison is fair."
> > >
> > > Aside from the text motivating the model, what do you think could be added or amended in this study to make it more clearly worthwhile of publishing going forward? We ask because you appear to be interested in the subject area.
> > >
> > > E.g.
> > > - One-shot learning for real applications is not sufficiently motivated?
> > > - Some experiments are missing (from your perspective)?
> > > - You think there are flaws in the model or comparison approach?
> > >
> > > Any further feedback would be highly appreciated.

---

### Official Review · AnonReviewer1 · 2018-11-05
**Unclear paper, difficult to understand how the algorithm works or why**

**Rating:** 3
**Confidence:** 1

**Review:**

The paper proposes a method whereby a neural network is trained and used as a data structure to assess approximate set membership. Unlike the Bloom filter, which uses hand-constructed hash functions to store data and a pre-specified method for answering queries, the Neural Bloom Filter learns both the Write function and the Read function (both are "soft" values rather than the hard binary values used in the Bloom filter). Experiments show that, when there is structure in the data set, the Neural Bloom Filter can achieve the same false positive rate with less space.

I had a hard time understanding how the model is trained. There is an encoding function, a write function, and a query function. The paper talks about one-shot meta-learning over a stream of data, but doesn't make it clear how those functions are learned. A lot of details are relegated to the Appendix. For instance B.2 talks about the encoder architecture for one of the experiments. But even that does not contain much detail, and it's not obvious how this is related to one-shot learning. Overall, the paper is written from the perspective of someone fully immersed in the details of the area, but who is unable to pop out of the details to explain to people who are not already familiar with the approach how it works. I would suggest rewriting to give an end-to-end picture of how it works, including details, without appendices. The approach sounds promising, but the exposition is not clear at all.

---

> ### Author Response · Authors · 2018-11-17
> **Explanation of training setup and why it's one-shot classification.**
>
> Thank you for reading the paper, and we apologize for its opacity upon first pass. We completely agree the paper has mis-judged its audience and was not easy to read straight-through, this feedback is very useful in correcting this. We wrote the paper for someone highly familiar with meta-learning memory-augmented neural networks but not familiar with bloom filters; this left out an important audience.
>
> --- Re. “I had a hard time understanding how the model is trained...”
>
> The model learns in one-shot because it observes a set S = (k1, k2, … kn) and writes it to a memory (or state) M with only one observation of this dataset. It then answers queries “is my query x in S” using the read operation, conditioning on the memory, M. It is the same one-shot classification approach as "Matching Networks" Vinyals et al. 2016 however we focus on classifying familiarity versus image or text class. We have added several paragraphs and an algorithm box with further explanation of the meta-learning training setup. We will just briefly summarize it here.
>
> We have a collection of sets Strain1, Strain2, , … Strainm reserved for training; and a collection of queries Q = {q1, q2, …, qL} and targets yi = 1 if qi in S and 0 otherwise. In the example of a database we can think of a given set Si = {k1, …, kN} as a set of rowkeys for a given file on disk (e.g. SSTable). We have many sets because we have many files; for training we have reserved some for an offline training routine.
>
> During training we calculate M = fwrite(S), and then we calculate oi = fread(S, qi). We calculate the cross-entropy loss L = \sumi yi log(oi) + (1 - yi)log(1-oi) and backprogate through the network (through the parameters controlling both the read, write, and encoder networks). One can consider the creation of M = fwrite(S) as a fast one-shot learning procedure; the network learns a state which can help it solve the classification problem, “is q in S?”. The slow-moving ‘meta-learning’ process is in the network parameters, which are slowly being optimized over several set membership tasks, i.e. several different sets S1:m, to be effective at one-shot classification. At test time, when we observe a new subset (or stream of elements) we can insert them with fwrite in one-shot and the resulting data-structure is the external memory, M.
>
> -- Re. “A lot of details are relegated to the Appendix. For instance B.2 talks about the encoder architecture for one of the experiments.”
>
> This is a good point. We have removed B. 2 from the appendix and promoted the details to the model section. Furthermore we have given an example instantiation of the full architecture in the model section, so one does not need to consult the appendix. We have not completely removed the appendix as some details are tangential discussion points (e.g. how to implement the model in sub-linear time) but other details, such as space comparison, are now described in more detail in the experiments section.
>
> We have significantly re-written the paper’s model and experiments section to remedy this --- please take a look and let us know if this addresses concerns.

---

### Author Response · Authors · 2018-11-17
**Substantial revision - thank you!**

Thank you for reading the paper and leaving your detailed feedback. The unified message from all three of you is that the paper could have done a better job in motivating the model, and describing the training regime. We have perhaps ‘regularized’ the paper’s contents too heavily in the endeavor to be succinct. We provide an updated manuscript with additions to ‘model’, ‘experiments’, and ‘related work’ --- an extra page of text. The proposed architecture is now better explained and the training regime is much more explicit. It’s a much better paper thanks to your comments.

If you think the problem setting has no potential for impact, or if you think there are fundamental flaws in our research approach then we would really appreciate feedback on this (and a rejection). Otherwise we would ask you to read the updated manuscript and update your response. We will also respond to each comment individually.

----

Key changes:

- Re-written ‘model’ section with a much clearer motivation. Added more specific details (e.g. encoder architecture) to model section, less reliance on appendix.

- Re-written experiments: explained meta-learning training in detail with algorithm box, explained why this is meta-learning / one-shot learning, added space comparison info, less reliance on appendix.

- Added speed comparison benchmarks (some peers were interested in us adding these numbers). The summary of these numbers is the latency of the neural bloom filter is much higher than a bloom filter, but the throughput can be comparable if the model is run on a gpu.

---

> ### Author Response · Authors · 2018-11-17
> **Re: "How is this different to Kraska et al. 2018"**
>
> One concern that reviewer 2 and 3 raised that we would like to quickly address is, ‘what is the point of this model vs kraska et al. 2018?’ The simple answer is that kraska et al. 2018 learns a set membership classifier by training a feed-forward neural classifier from scratch over many (hundreds to thousands) epochs of the storage set S, and it is compressed into the weights of the network. We propose a method where a neural network learns to produce a classifier with a single pass over S, and the set is represented by an external memory M of compressed activations.
>
> In the case of a banned URL list, where S may not change very much, it may be tenable to use the kraska et al. 2018 approach with multiple epochs of gradient descent. In the case of databases that uses Bloom Filters (e.g. Google Bigtable, Apache Cassandra, Redis) where one may have thousands of separate bloom filters (one per disk file, say) which are dynamically updating, it is impractical to train thousands of separate networks from scratch. Thus a one-shot approach (our paper) is absolutely necessary, and this paper serves as an existence proof that significant compression can be obtained in this challenging setting.

---

### Author Response · Authors · 2018-11-21
**Feedback on revision**

Dear reviewers,

Given the consensus was that the model motivation and the one-shot learning setup was not clear enough, versus fundamental disagreements with the subject area or potential impact, it would be very valuable for feedback on the paper revision.  The principal changes are found in Section 3, Section 5 intro (+ Algorithm 1) & Section 5.1,  box. We have surveyed several of our peers unfamiliar with meta-learning and they said they understood the training regime much better and felt 80%+ sure of how the model was trained and evaluated. So we would be very grateful if you could consider our response and paper revision!

---

### Author Response · Authors · 2018-11-24
**Updated speed benchmark with LSTM**

To supplement the discussion on "why create this model versus use an LSTM or variant". Aside from the fact that we could not get any RNN to solve the database task with 5,000 elements; we ran an LSTM on the speed benchmark for this task to determine how much slower/faster it would be ... If we could somehow train it to solve this task.

Insertion throughput for Bloom Filter on: ~60K (CPU)
Insertion throughput for Neural Bloom Filter: ~4K (CPU), ~58K (GPU)
Insertion throughput for LSTM: ~2K (CPU), ~5K (GPU).

So as discussed earlier, a Neural Bloom Filter can match the throughput of an Bloom Filter when run on a GPU. A couple of GPUs reserved along with the thousand of CPU cores for a large Bigtable database seems feasible. However if we look at the LSTM numbers, the insertion throughput is about 10x less (5K insertions per sec vs 60K). Thus the sequential write scheme of RNNs (in this case, an LSTM) is not only a problem from an optimization-perspective - as the LSTM fails to learn the task - but also reduced the throughput of insertions by an order of magnitude during evaluation. We have added these numbers to the Appendix G and discussed this point in the main paper, Section 5.4. This is just extra empirical evidence that there is room for a memory model with a feed-forward & compressive write scheme, alike to a bloom filter.

---

### Meta-Review · Area_Chair1 · 2018-12-17
**Interesting approach, empirical validation needs strengthening**

**Confidence:** 4
**Recommendation:** Reject

**Metareview:**

This work proposes and interesting approach to learn approximate set membership. While the proposed architecture is rather closely related to existing work, it is still interesting, as recognized by reviewers. Authors's substantial rewrites has also helped make the paper clearer. However, the empirical merits of the approach are still a bit limited; when combined with the narrow novelty compared to existing work, this makes the overall contribution a bit too thin for ICLR. Authors are encouraged to strengthen their work by showing more convincing practical benefit of their approach.